# Phylomitogenomics bolsters the high-level classification of Demospongiae (phylum Porifera)

**Dennis V. Lavrov**[1]*, **Maria C. Diaz**[2,3], **Manuel Maldonado**[4], **Christine C. Morrow**[5,6], **Thierry Perez**[7], **Shirley A. Pomponi**[2], **Robert W. Thacker**[8,9]

**1** Department of Ecology, Evolution, and Organismal Biology, Iowa State University, Ames, Iowa, United States of America, **2** Harbor Branch Oceanographic Institute, Florida Atlantic University, Fort Pierce, Florida, United States of America, **3** Museo Marino de Margarita, Boca de Río, Nueva Esparta, Venezuela, **4** Department of Marine Ecology, Centro de Estudios Avanzados de Blanes (CEAB-CSIC), Girona, Spain, **5** Zoology Department, School of Natural Sciences & Ryan Institute, NUI Galway, University Road, Galway, Ireland, **6** Ireland and Queen's University Marine Laboratory, Portaferry, Northern Ireland, **7** Institut Méditerranéen de la Biodiversité et d'Ecologie marine et continentale (IMBE), CNRS, Aix-Marseille Université, IRD, Avignon Université City, Provence, France, **8** Department of Ecology and Evolution, Stony Brook University, Stony Brook, NY, United States of America, **9** Smithsonian Tropical Research Institute, Balboa, Panama City, Republic of Panama

* dlavrov@iastate.edu

**Data Availability Statement:** All newly sequenced and/or assembled mtDNA sequences were deposited to the Genbank under accession

## Abstract

Class Demospongiae is the largest in the phylum Porifera (Sponges) and encompasses nearly 8,000 accepted species in three subclasses: Keratosa, Verongimorpha, and Heteroscleromorpha. Subclass Heteroscleromorpha contains ∼90% of demosponge species and is subdivided into 17 orders. The higher level classification of demosponges underwent major revision as the result of nearly three decades of molecular studies. However, because most of the previous molecular work only utilized partial data from a small number of nuclear and mitochondrial (mt) genes, this classification scheme needs to be tested by larger datasets. Here we compiled a mt dataset for 136 demosponge species—including 64 complete or nearly complete and six partial mt-genome sequences determined or assembled for this study—and used it to test phylogenetic relationships among Demospongiae in general and Heteroscleromorpha in particular. We also investigated the phylogenetic position of *Myceliospongia araneosa*, a highly unusual demosponge without spicules and spongin fibers, currently classified as Demospongiae *incertae sedis*, for which molecular data were not available. Our results support the previously inferred sister-group relationship between Heteroscleromorpha and Keratosa + Verongimorpha and suggest five main clades within Heteroscleromorpha: Clade C0 composed of order Haplosclerida; Clade C1 composed of Scopalinida, Sphaerocladina, and Spongillida; Clade C2 composed of Axinellida, Biemnida, Bubarida; Clade C3 composed of Tetractinellida; and Clade C4 composed of Agelasida, Clionaida, Desmacellida, Merliida, Suberitida, Poecilosclerida, Polymastiida, and Tethyida. The inferred relationships among these clades were (C0(C1(C2(C3+C4)))). Analysis of molecular data from *M. araneosa* placed it in the C3 clade as a sister taxon to the highly skeletonized tetractinellids *Microscleroderma* sp. and *Leiodermatium* sp. Molecular clock analysis dated divergences among the major clades in Heteroscleromorpha from the

numbers OM729606-OM729671, All trees and alignments are available at the website https://lavrovlab.github.io/Demosponge-phylogeny.

**Funding:** This work was supported by the National Science Foundation's Assembling the Tree of Life program (DEB No. 0829783 to DVL, DEB No. 0829986 to RWT, as well as DEB awards 0829763, 0829791), grant BFU2008-00227/BMC and PID2019-108627RB-I00 of the Spanish Ministry of Sciences and Innovation to MM, and by internal funds from Iowa State University. In addition, we gratefully acknowledge postdoctoral funding from the Irish Research Council to CCM. The funders had no role in study design, data collection and analysis, decision to publish, or preparation of the manuscript.

**Competing interests:** The authors have declared that no competing interests exist.

Cambrian to the Early Silurian, the origins of most heteroscleromorph orders in the middle Paleozoic, and the most basal splits within these orders around the Paleozoic to Mesozoic transition. Overall, the results of this study are mostly congruent with the accepted classification of Heteroscleromorpha, but add temporal perspective and new resolution to phylogenetic relationships within this subclass.

## Introduction

The phylum Porifera (sponges) consists of four taxonomic classes: Demospongiae, Homoscleromorpha, Hexactinellida, and Calcarea [1]. Among them, class Demospongiae Sollas 1895 is by far the largest (>82% of accepted species) and morphologically the most diverse [2]. Demosponges are found in both freshwater and marine environments from intertidal zone to abyssal depth and include familiar commercial sponges [3] as well as such oddities as carnivorous sponges [4]. Demosponges fulfill several important roles in benthic ecosystems, being essential players both in the carbon flux [5] and in the silicon cycle [6]. In addition, sponges have the capacity to modify boundary flow as they pump large volumes of seawater into the water column [7]. With the decline of reef-building corals on tropical reefs, sponges are becoming one of the most important structural elements in these ecosystems and provide shelter to a variety of other species [8, 9]. From an evolutionary perspective, sponges –– one of the two main candidates for being the sister group to the rest of the animals [10, 11] –– provide insight into the common ancestor of all animals [12, 13]. This knowledge, in turn, can improve our understanding of the origin of animal multicellularity and evolution of animal body plans (reviewed in [14, 15]). Indeed, several genomic [12, 16–19] and transcriptomic [13, 20, 21] studies of demosponges have been used to infer steps in animal evolution as well as to clarify various aspects of sponge biology.

The relationships among the higher taxa of demosponges have been studied since the second half of the 19th century but are still only partially resolved (reviewed in [22]). The most recent pre-molecular taxonomic treatment of the phylum Porifera, *Systema Porifera* [23], subdivided demosponges into three subclasses, and 14 orders, but warned that "resolving the higher systematics of sponges is clearly beyond the capabilities of this present book" [24]. The advent of molecular systematics led to the rejection of many higher-level taxa defined based on morphological and embryological data and to the recognition of four major lineages within the class: Keratosa (G1) (Dictyoceratida + Dendroceratida), Verongimorpha (G2) (Chondrosida, Halisarcida, and Verongida), Marine Haplosclerida (G3), and the remaining orders (G4) (at the time, Agelasida, Hadromerida, Halichondrida, Tetractinellida (Astrophorina + Spirophorina), Poecilosclerida, and Spongillina (freshwater Haplosclerida)) [25–28]. In addition, a confluence of ultrastructural, embryological and molecular studies resulted in removal of the former sub-class Homoscleromorpha Lévi, 1973 from Demospongiae to form the fourth class of Porifera [1].

A revised classification of the Demospongiae was proposed in 2015 that united G3 and G4 into the subclass Heteroscleromorpha and subdivided the latter group into 17 orders [29]. However, it left phylogenetic relationships among these orders mostly unresolved. Furthermore, because the new system—now accepted as the framework for the demosponge classification in the World Porifera Database http://www.marinespecies.org/porifera/index.php—was based primarily on partial nuclear rRNA and mitochondrial *cox1* data, it needed to be tested with larger datasets.

Mitochondrial genomes (mt-genomes) provide two primary data types for phylogenetic inference: DNA sequence and gene arrangements. Mt-sequence data (often translated and concatenated coding sequences) have been used extensively in molecular phylogenetics [30–34] and are particularly well suited for the analysis of demosponge relationships because of the low rate of evolution and relatively homogeneous composition of mtDNA sequences in this group [26, 35–38]. Mt-gene arrangement data have also been used both for the reconstruction of global animal relationships [26, 35] and for testing specific phylogenetic hypotheses (*e.g.*, [39]). However, this dataset provides fewer characters for phylogenetic inference and its analysis is computationally more challenging, so the latter application is more common.

As part of the Porifera Tree of Life project https://portol.org/, we determined mtDNA sequences from 64 demosponges and assembled six more from available genomic and transcriptomic data. Here we report these data and use them, along with publicly available mt-genomes to reconstruct and test demosponge phylogenetic relationships. Importantly, we included in our dataset *Myceliospongia araneosa* Vacelet & Pérez, 1998 (currently classified as Demospongiae *incertae sedis*) and resolved its phylogenetic position. In addition, we tested the phylogenetic affinities of several demosponge species used for genomic projects and conducted a molecular clock analysis of demosponge evolution.

## Materials and methods

### Data collection

**Overview.** For this project, we PCR amplified, sequenced, assembled, and annotated complete or nearly complete mtDNA sequences from 57 species of demosponges including 48 belonging to the subclass Heteroscleromorpha. Two additional mt-genome sequences were assembled from low coverage Illumina DNAseq data generated for this project. Furthermore we assembled and annotated five complete mtDNA sequences from publicly available DNAseq and RNAseq data (Table 1). Combined with 61 previously published mt-genomes, we compiled a dataset of 125 complete and nearly complete mt-genomes https://lavrovlab.github.io/Demosponge-phylogeny/published.html. To this dataset we added partial mtDNA sequences from one species of Merliida and one species of Desmacellida as well as a partial mtDNA sequence from an unknown species most closely related to *Plenaster craigi*. In addition, we utilized available RNAseq data from *Scopalina* sp. CDV-2016 and newly generated DNAseq data from *Hymerhabdia typica* and *Mycale escarlatei* to assemble most of the mitochondrial coding sequences for these species. Finally, we used mt-coding sequences from five incomplete mt-genomes reported by Plese *et al.* (2021), resulting in a final dataset of 136 taxa.

**Taxon sampling.** Species used in this study were chosen to cover much of the suprafamilial diversity in Heteroscleromorpha (Table 1). Eleven additional species of demosponges outside of this subclass were sampled and included in the analyses as outgroups (Table 2). Tissue subsamples, and/or DNA aliquots were derived from three main sources: (1) PorToL-supported expeditions to the Smithsonian Tropical Research Institute at Bocas del Toro, Panama, in 2009 and 2012, where subsamples of specimens were stored in 3M Guanidinium Chloride solution; (2) the Moorea Biocode project, Moorea, French Polynesia (https://geome-db.org/workbench/project-overview?projectId=75), where subsamples of specimens were stored in 3M Guanidinium Chloride solution, and (3) collection effort by C.C.M with the Ulster Museum, Belfast with samples stored in 94% ethanol. Vouchers for these samples were deposited to the the National Museum of Natural History (Washington, USA) (sample IDs starting with USNM and BMOO) and the Belfast Ulster Museum (Belfast, Northern Ireland) (BELUM MC). Additional specimens were collected by M.M. from Blanes coast (41°40.21'N; 2°48.14'E) (sample IDs starting with BL) or were subsampled from the Florida Atlantic University Harbor

**Table 1. Heteroscleromorph species for which mt-genomes were assembled in this study.**

| Order | Species | Sample ID | Genbank # | Seq. technology | Location/Source |
|---|---|---|---|---|---|
| Haplosclerida | *Chalinula loosanoffi* (Hartman, 1958) | CLOOS | OM729611 | Illumina | Virginia, USA |
| –//– | *Haliclona (Reniera) implexiformis* (Hechtel, 1965) | USNM 1133760 | OM729617 | Sanger | Panama |
| –//– | *Haliclona (Rhizoniera) indistincta* (Bowerbank, 1866) | BELUM MC7978 | OM729618 | Sanger | Celtic Sea |
| –//– | *Haliclona (Reniera) manglaris* Alcolado, 1984 | USNM 1133711 | OM729634 | Illumina | Panama |
| –//– | *Haliclona poecillastroides* (Vacelet, 1969) | MRS1163 | OM729629 | Illumina | French Mediterranean |
| –//– | *Neopetrosia sigmafera* Vicente, Ríos, Zea & Toonen, 2019 | USNM 1133752 | OM729643 | Illumina | Panama |
| –//– | *Niphates digitalis* (Lamarck, 1814) | FL06507 | OM729631 | Sanger | Florida |
| –//– | *Niphates erecta* Duchassaing & Michelotti, 1864 | USNM 1133708 | OM729633 | Illumina | Panama |
| –//– | *Xestospongia testudinaria* (Lamarck, 1815) | – | OM729660 | Assembly | SRR1736373 |
| Suberitida | *Halichondria (Halichondria) panicea* (Pallas, 1766) | BELUM MC7975 | OM729622 | Illumina | Celtic Sea |
| –//– | *Hymeniacidon perlevis* (Montagu, 1814) | BELUM MC7974 | OM729620 | Illumina | Celtic Sea |
| –//– | *Suberites ficus* (Johnston, 1842) | BELUM MC7979 | OM729652 | Illumina | Celtic Sea |
| Polymastiida | *Polymastia tenax* Pulitzer-Finali, 1986 | USNM 1133747 | OM729642 | Illumina | Panama |
| Tethyida | *Adreus fascicularis* (Bowerbank, 1866) | BELUM MC6778 | OM729607 | Illumina | Celtic Sea |
| –//– | *Tethya wilhelma* Sarà, Sarà, Nickel & Brümmer, 2001 | TW | OM729659 | Sanger | Wilhelma Aquarium, Stuttgart |
| Clionaida | *Cliona varians* (Duchassaing & Michelotti, 1864) | HBOM 19-I-09–1–2 | OM729613 | Illumina | Florida Keys |
| –//– | *Clionaopsis* sp. | BMOO_16278 | OM729624 | Illumina | Moorea |
| –//– | *Diplastrella bistellata* (Schmidt, 1862) | BL03–2011 | OM729614 | Illumina | Blanes coast, Spanish Mediterranean |
| –//– | *Placospongia intermedia* Sollas, 1888 | USNM 1133726 | OM729639 | Illumina | Panama |
| –//– | *Spirastrella cunctatrix* Schmidt, 1868 | BL02–2011 | OM729627 | Illumina | Spanish Mediterranean |
| Poecilosclerida | Chondropsidae sp. | BMOO_16298 | OM729625 | Illumina | Moorea |
| –//– | *Clathria (Thalysias) curacaoensis* Arndt, 1927 | USNM 1133715 | OM729636 | Illumina | Panama |
| –//– | *Crambe crambe* (Schmidt, 1862) | BL02–2007 | OM729612 | Sanger | Spanish Mediterranean |
| –//– | *Hymedesmia (Stylopus)* sp. | BMOO_16335 | OM729626 | Illumina | Moorea |
| –//– | *Lissodendoryx colombiensis* Zea & van Soest, 1986 | USNM 1133712 | OM729635 | Illumina | Panama |
| –//– | *Phorbas* sp. | – | OM729630 | gDNA Illumina | French Mediterranean |
| –//– | *Phorbas amaranthus* Duchassaing & Michelotti, 1864 | USNM 1133787 | OM729646 | Illumina | Panama |
| –//– | *Tedania (Tedania) ignis* (Duchassaing & Michelotti, 1864) | USNM 1133763 | OM729644 | Illumina+454 | Panama |
| Agelasida | *Astrosclera willeyana* Lister, 1900 | GW1144 | OM729609 | Illumina | GBR, Australia |
| –//– | *Stylissa carteri* (Dendy, 1889) | – | OM729650 | Assembly | SRR1738070 |
| Axinellida | *Raspaciona aculeata* (Johnston, 1842) | BL08–2007 | OM729649 | Sanger | Spanish Mediterranean |
| –//– | *Stelligera stuposa* (Ellis & Solander, 1786) | BELUM MC8291 | OM729655 | Illumina | Celtic Sea |
| –//– | *Heteroxya beauforti* Morrow, 2019 | BELUM MC7794 | OM729610 | Illumina | Celtic Sea |
| –//– | *Axinella infundibuliformis* (Linnaeus, 1759) | BELUM MC8292 | OM729623 | Illumina | Celtic Sea |
| –//– | *Axinella polypoides* Schmidt, 1862 | BELUM MC7308 | OM729608 | Illumina | Celtic Sea |
| Bubarida | *Acanthella acuta* Schmidt, 1862 | BL04–2011 | OM729606 | Illumina | Spanish Mediterranean |
| –//– | *Dictyonella marsilii* (Topsent, 1893) | BL06–2011 | OM729628 | Illumina | Spanish Mediterranean |
| –//– | Dictyonellidae sp. | USNM 1133716 | OM729637 | Illumina | Panama |
| –//– | *Phakellia ventilabrum* (Linnaeus, 1767) | BELUM MC8294 | OM729648 | Illumina | Celtic Sea |
| –//– | *Svenzea flava** (Lehnert & van Soest, 1999) | P12–394 | OM729653 | Sanger | Panama |
| ??? | *Petromica* sp.* | HBOM 003:02026 | OM729647 | Sanger | US Gulf of Mexico |
| Tetractinellida | *Dercitus (Halinastra) luteus* (Pulitzer-Finali, 1986) | HBOM 003:02024 | OM729616 | Illumina | Guanaja, Honduras |
| –//– | *Leiodermatium* sp. | HBOM 003:02027 | OM729658 | Illumina | Florida, Miami Terrace |
| –//– | *Microscleroderma* sp. | USNM 1133739 | OM729641 | Illumina | Panama |
| –//– | *Myceliospongia araneosa* Vacelet & Pérez, 1998 | MRS1151 | OM729621 | Illumina | French Mediterranean |

*(Continued)*

**Table 1.** (Continued)

| Order | Species | Sample ID | Genbank # | Seq. technology | Location/Source |
|---|---|---|---|---|---|
| –//– | *Neophrissospongia* sp. | HBOM 28-V-91–2–001 | OM729632 | Illumina | Madeira Archipelago |
| –//– | *Stelletta fibrosa* (Schmidt, 1870) | USNM 1133730 | OM729640 | Illumina | Panama |
| Biemnida | *Biemna caribea* Pulitzer-Finali, 1986 | USNM 1133766 | OM729645 | 454 | Panama |
| –//– | *Neofibularia nolitangere* (Duchassaing & Michelotti, 1864) | USNM 1133723 | OM729638 | 454 | Panama |
| Spongillida | *Spongilla lacustris* (Linnaeus, 1759) | – | OM729654 | Assembly | SRR1168575 |
| Scopalinida | *Scopalina* sp. | BELUM MC7158 | OM729651 | gDNA Illumina | Celtic Sea |
| –//– | *Svenzea zeai* (Alvarez, van Soest & Rützler, 1998) | USNM 1133762 | OM729656 | Sanger | Panama |
| Sphaerocladina | *Vetulina* sp. | HBOM 003:02025 | OM729657 | 454 | Curacao |
| Merliida | *Hamacantha (Hamacantha) johnsoni*** (Bowerbank, 1864) | BELUM MC2019.3 | OM729619 | Illumina | Celtic Sea |
| Desmacellida | *Desmacella informis*** (Stephens, 1916) | BELUM MC2019.1 | OM729615 | Illumina | Celtic Sea |

Additional information for samples collected from Panama (vouchers USNM #) is available at https://collections.nmnh.si.edu/search/iz. Additional information for samples collected in Moorea (vouchers BMOO_#) is available at https://geome-db.org/query. Phylogenetic position of species marked with an asterisk (*) does not correspond to their classification. Two asterisks (**) indicate partial mt-genomes.

Branch Oceanographic Institute collection (sample IDs starting with HBOM). *Myceliospongia araneosa* was collected by T.P. from the "3PP" cave near La Ciotat (43°09.47'N—05°36.01'E) in the Mediterranean Sea. *Thymosia* sp. was collected by M.M. at the Chafarinas Islands (35°11.05'N; 2°26.08'E) in the Mediterranean Sea. Further samples were provided by Steven Cook (SDCC-NZ-363), Alexander Ereskovsky (MRS0816 and MRS1163), April Hill (CLOOS), Gisele Lôbo-Hajdu (GLH1203), Michael Nickel (TW), Julie Reveillaud (HPRUV), and Gert Wörheide (GW960 and GW1144). The Research activities at the Bocas del Toro Research Station of the Smithsonian Tropical Research Institute in Panama and export of biological materials were conducted with permission of the Autoridad de los Recursos Acuaticos de Panama. No permits were required to sample other specimens.

**DNA extraction, PCR amplification.** Collected sponge samples were preserved in either 95% ethanol or 3M Guanidinium Chloride solution. Total DNA was extracted with a phenol-chloroform method modified from [40]. Porifera-optimized conserved primers developed in our laboratory [26] were used to amplify short (400–1000 nucleotide) fragments of several

**Table 2. Other demosponge species for which mt-genomes were assembled in this study.**

| Order | Species | Sample ID | Genbank # | Sequencing technology | Location/Source |
|---|---|---|---|---|---|
| Dendroceratida | *Darwinella gardineri* Topsent, 1905 | MRS0816 | OM729666 | Sanger + Illumina | French Mediterranean |
| –//– | *Dictyodendrilla dendyi* Bergquist, 1996 | SDCC-NZ-363 | OM729664 | Sanger | New Zealand |
| Dictyoceratida | *Dysidea etheria* de Laubenfels, 1936 | FL06501 | OM729665 | Sanger | Florida Keys |
| –//– | *Pleraplysilla spinifera* (Schulze, 1879) | – | OM729671 | RNAseq assembly | SRR3417588 |
| –//– | *Phyllospongia foliascens* (Pallas, 1766) | GW960 | OM729661 | Illumina | Lizard Island, Australia |
| Chondrosiida | *Chondrosia reniformis* Nardo, 1847 | BL01–2007 | OM729663 | Sanger | Spanish Mediterranean |
| Chondrillida | *Thymosia* sp. | BL09–2011 | OM729662 | Illumina | Morocco |
| –//– | *Halisarca caerulea* Vacelet & Donadey, 1987 | – | OM729667 | RNAseq assembly | SRR5234759 |
| Verongida | *Aiolochroia crassa* (Hyatt, 1875) | USNM 1133710 | OM729670 | Illumina | Panama |
| –//– | *Hexadella pruvoti* Topsent, 1896 | HPRUV | OM729668 | Illumina | French Mediterranean |
| –//– | Pseudoceratinidae sp. | BMOO_16215 | OM729669 | Illumina | Moorea |

mitochondrial genes for each species. Two species-specific primers were designed for each of these genes for PCR amplification. Complete mtDNA was amplified in several overlapping fragments using the Long and Accurate (LA) PCR kit from TAKARA.

**Sequencing.**　Three sequencing technologies were utilized in the project (Tables 1 and 2). MtDNA sequences from 11 species were determined using the Sanger method [41]. For each of these species all LA-PCR fragments were combined in equimolar concentrations, sheared into pieces 1–2 kb in size and cloned using the TOPO Shotgun Subcloning Kit from Invitrogen. Colonies containing inserts were collected, grown overnight in 96-well blocks and submitted to the DNA Sequencing and Synthesis Facility of the ISU Office of Biotechnology for high-throughput plasmid preparation and sequencing on the facility's Applied Biosystems 3730xl DNA Analyzer. Gaps in the assembly were filled by primer-walking.

MtDNA sequences from three species were determined using 454's sequencing technology. PCR reactions for each species were combined in equimolar concentration, sheared and barcoded as described in Gazave et al. [42]. Barcoded PCR fragments were combined together and used for the GS FLX Titanium library preparation (454 Life Sciences). Pyrosequencing was carried out on a Genome Sequencer FLX Instrument (454 Life Sciences) at the University of Indiana Center for Genomics and Bioinformatics.

Finally, mtDNA sequences for 46 species were determined by using Illumina technology. For these species, PCR reactions for each species were combined in equimolar concentration, sheared and combined together with or without barcoding. Libraries were prepared using the Illumina TruSeq DNA PCR-Free Library Prep Kits. Sequencing was carried on MiSeq and HiSeq instruments at the DNA Sequencing and Synthesis Facility of the ISU Office of Biotechnology.

**Sequence assembly.**　Different assemblers were used depending on the type of data collected. The STADEN package v. 1.6.0 [43] with Phred basecaller [44, 45] was used to assemble the Sanger sequences. Abyss [46], Mira [47], PCAP [48], and SPAdes [49] were used to assemble 454 and Illumina sequences. In most cases, several programs were utilized for the assembly and results compared and compiled together. When barcodes were used, sequences were first separated by the barcode. When barcodes were not used, species selection was carried out to exclude closely related species from the same library. In the latter case, assembled sequences were identified using short sequences generated for primer design. PCR and Sanger sequencing was used to resolve any ambiguities.

**Sequence annotation.**　We used flip v. 2.1.1 (http://megasun.bch.umontreal.ca/ogmp/ogmpid.html) to predict ORFs in assembled sequences; similarity searches in local databases and in GenBank using FASTA [50] and NCBI BLAST network service [51], respectively, to identify them. Protein-coding genes were aligned with their homologues from other species and their 5' and 3' ends inspected for alternative start and stop codons. Genes for small and large subunit ribosomal RNAs (*rns* and *rnl*, respectively) were identified based on their similarity to homologous genes in other species, and their 5' and 3' ends were predicted based on sequence and secondary structure conservation. Transfer RNA genes were identified by the tRNAscan-SE program [52]. RNAweasel [53] was used to search for introns in the coding sequences. The exact positions of introns were adjusted based on alignments of coding sequences that contained them.

## Phylogenetic inference

**Phylogenetic analysis based on mitochondrial coding sequences.**　Two datasets were constructed for phylogenetic analysis based on mitochondrial coding sequences. The first dataset comprised 136 demosponge species for which we had complete/nearly complete

mtDNA data or most individual mt-coding sequences (CDS). Inferred amino acid sequences of individual mitochondrial proteins were aligned with Mafft v7.475 [54] using L-INS-i stategy. Conserved blocks within the alignments were selected with Gblocks 0.91b [55] using relaxed parameters (parameters 1 and 2 = 0.5, parameter 3 = 8, parameter 4 = 5, all gap positions in parameter 5). Cleaned alignments were concatenated into a supermatrix containing 3,634 amino acid positions for 136 species. The second dataset comprised the same 136 demosponge species plus nine additional species of Homoscleromorpha and was constructed as the first dataset, except the concatenated alignment was filtered with CD-Hit [56] to remove sequences with >95% identity. The final alignment for the second dataset contained 83 species and 3,633 amino acid positions. Both datasets were analyzed with PhyloBayes MPI 1.9 [57] under the CAT+GTR model (-cat -gtr). The chains were sampled every 10th tree after the first 1000 burn-in cycles to calculate consensus trees.

**Gene order analysis.** Mitochondrial gene orders were converted to gene adjacency matrices using the *gogo* program (https://github.com/dlavrov/bio-geneorder, unpublished). The matrices were further modified as required by TNT and RAxML and used in these programs to infer the Maximum Parsimony (MP) and Maximum Likelihood (ML) trees, accordingly. For the parsimony analysis, we tried both the traditional (*i.e.*, random addition of sequences + TBR branch swapping followed by additional branch swapping of trees in memory) and "new technology" (*i.e.*, with Ratchet, Tree-Drifting, and Tree-Fusing followed by additional branch swapping) strategies implemented in TNT [58]. Run scripts for the TNT analysis are available in the supplementary GitHub repository. For the ML analysis, we used the multistate model in RAxML-NG v. 1.1.0 [59]: "raxml-ng - -all - -msa 77taxa.phy - -model MUL-TI13_MK." To check for the effect of more frequent tRNA rearrangements in animal mitochondrial genomes, we created an alternative adjacency matrix based, where position of each gene was recorded relative to the position of the closest major (*i.e.*, protein or rRNA) upstream and downstream genes and repeated the MP and ML analyses.

## Molecular clock analysis

PhyloBayes 4.1c [60] was used for the molecular clock analysis with the fixed tree topology inferred from mitochondrial coding sequences. The model of sequence evolution was the same as for unconstrained phylogenetic analysis: a generalized time-reversible (GTR) amino acid substitution matrix (-gtr), a Dirichlet mixture profile (-cat), and a discrete gamma distribution with four categories -dgam [4]. We used the log normal (Brownian) autocorrelated clock [61] (-ln) model for the analysis and ran two chains for >15,000 cycles. Convergence was assessed by estimating discrepancies and effective sizes for continuous variables in the model using *tracecomp* with 250 generations removed as burn-in ('tracecomp -x 250'). Three calibration points were utilized for the analysis: A uniform prior between 541 and 515 MA (beginning of Cambrian—crown-group heteroscleromorph fossil [62]) was placed on the root of the demosponges (split between Keratosa+Verongimorpha and Heteroscleromorpha). The split between Spongillida and *Vetulina* sp. was constrained between 410 and 298 MY (the lower bound is defined by the observation that species diversification in lakes prior to the Devonian was limited by low nutrient loads and high sediment loads [63], the upper bound is defined by the first reported freshwater sponge fossil [64]). The origin of the crown group Baikal sponges (the split between *Baikalospongia intermedia profundalis* and *Lubomirskia baikalensis*) between 30 and 6 MY based on Lake Baikal history and fossil record of Lake Baikal sponges [65]. All calibration ranges were specified as soft bounds (-sb option), which allocates 0.025 of the total probability outside the specified bounds. Dates were assessed by running *readdiv* with 250 generations removed as burn-in and every 10th generation sampled for each

analysis ('readdiv -x 250 10'). The chain with greater number of points was utilized for each molecular clock method.

## Results

### Mt-genome organization

**Structure and gene content.** Most newly characterized mt-genomes were circular-mapping molecules, each containing a conserved set of 14 protein-coding, two ribosomal RNA (rRNA) and 24 or 25 tRNA genes (Fig 1). However, we found the following exceptions to this typical organization:

1. The mitochondrial genome of the poecilosclerid *Mycale escarlatei* did not assemble into a single circular molecule. Instead, several alternative arrangements have been found for most mitochondrial genes, indicating an unusual and likely multi-chromosomal genome architecture. Preliminary data from several other species in the genus *Mycale* suggest a similar organization (unpublished).

2. The mitochondrial genome of the *Scopalina* sp. assembled into three contigs with AT-rich sequences at the ends of each of them. The contig containing *cox1* had twice the coverage of the other two. It is not clear whether these contigs represent individual chromosomes or are results of a genome duplication/mis-assembly.

3. *atp9* was not identified in mt-genomes of *Lissodendoryx* sp., Chondropsidae sp. MO1046 (Poecilosclerida) and *Neopetrosia sigmafera* (Haplosclerida). Both *atp9* and *atp8* were missing in those of *Niphates digitalis* and *N. erecta* (Haplosclerida).

4. No stop codon was identified in *nad3* of *Lissodendoryx* sp. and closely related *Tedania ignis*. Furthermore, this gene was immediately followed by in-frame *nad4L*, suggesting that the two genes are fused in these species.

5. Loss of multiple mt-tRNA genes was inferred in the two *Niphates* species. In addition, we observed:

    (a). a loss of *trnC(gca)* in *Crambe crambe*;

    (b). a loss of *trnK(uuu)* in *Neopetrosia sigmafera*;

    (c). losses of *trnD(guc)* in *Agelas schmidtii* and closely related *Astroclera willeyana*, as well as *Clathria curacaoensis*;

    (d). putative losses of *trnP(ugg)* and *trnL(uag)* in *Scopalina* spp;
    Outside of Heteroscleromorpha, we observed losses of multiple tRNA genes in *Chondrosia reniformis* (Chondrosiidae, Verongimorpha) and Keratosa.

6. We also observed a few unusual and/or redundant tRNAs in newly sequenced mt-genomes:

    (a). *trnI(aau)* instead of the usual *trnI(gau)* was found in *Adreus fascicularis*;

    (b). *trnR(acg)* instead of *trnR(ucg)* was found in *Cliona varians*;

    (c). *trnL(caa)*, a third gene for Leucine tRNA was found in *Heteroxya beauforti*;

    (d). *trnY(aua)*, in addition to *trnY(gua)*, was found in *Negombata magnifica*;

    (e). unusual *trnX(uua)* that would be predicted to read the stop codon UAA was found in *Stelligera stuposa* but had the lowest cove score among all tRNAs in this species;

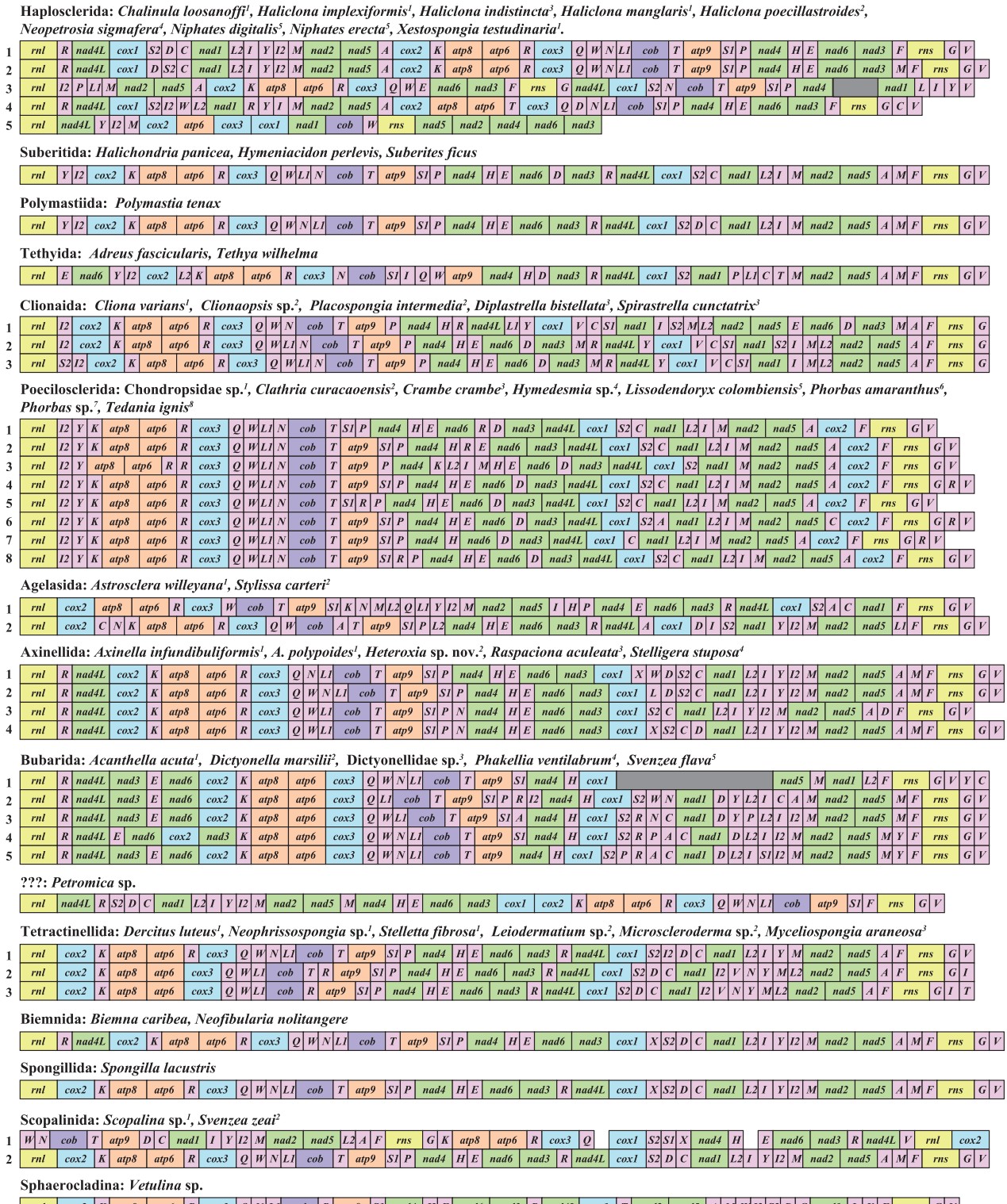

**Fig 1. Gene arrangements in heteroscleromorph mt-genomes determined for this study.** Superscript number associated with each species name refers to a unique gene order marked by the same number. Species are grouped mainly by taxonomic orders. In a few cases when taxonomy of a species is inconsistent with the results of phylogenetic analyses, the species is grouped according to the latter results. Because phylogenetic analysis does not place *Petromica* sp. in any accepted orders of demosponges, its systematic position is marked as "???". Protein and rRNA genes (larger boxes) are: *atp6*, *8–9*—subunits 6, 8 and 9 of the F0 ATPase, *cox1–3*—cytochrome *c* oxidase subunits 1–3, *cob*—apocytochrome *b* (*cob*), *nad1–6* and

*nad4L*—NADH dehydrogenase subunits 1–6 and 4L, *rns* and *rnl*—small and large subunit rRNAs. tRNA genes (smaller boxes) are abbreviated using the one-letter amino acid code. The two arginine, isoleucine, leucine, and serine tRNA genes are differentiated by numbers with *trnR(ucg)* marked as *R1*, *trnR(ucu)*—as *R2*, *trnI(gau)*—as *I1*, *trnI(cau)*—as *I2*, *trnL(uag)*—as *L1*, *trnL(uaa)* as *L2*, *trnS(ucu)*—as *S1*, and *trnS(uga)*—as *S2*. All genes are transcribed from left to right. Genes are not drawn to scale and intergenic regions are not shown. Missing sequences are indicated by gray boxes.

(f).  *trnA(ugc)* was present twice in *Stylissa carteri* as in closely related *Axinella corrugata*, where it was shown to be recruited from *trnT(ugu)* [66];

(g).  *trnR(ucu)* was present twice in *Vetulina* sp.

Most new demosponge mitochondrial genomes were in 16–22 kbp size range, with a mean size of ∼ 20.8 kbp. A few mitochondrial genomes were larger in size (>30kpb in *Dysidea etheria*), mostly due to the expansions of non-coding regions. All analyzed mitochondrial genomes had similar nucleotide composition (A+T content between 56–74%) and, with two exceptions, displayed overall negative AT- and positive GC-skews of the coding strand (the two exceptions were *Dysidea etheria* with AT-skew = 0.01 and *Scopalina* sp., with AT-skew = 0.03).

**Gene order.**   Mt-genomes of newly characterized demosponges shared between 0 to 41 gene boundaries. The largest differences, as expected, occurred among subclasses of demosponges and are exacerbated by the loss of the majority of tRNA genes in Keratosa, but also *Chondrosia reniformis* and the two *Niphates* species. Most of the differences in mitochondrial gene orders were caused by transpositions of tRNA genes. However, rearrangements of "major" (protein and rRNA) genes were also present. Furthermore, *Aiolochroia crassa* and the underscribed Pseudoceratinidae species MO1014 had the same inversion in mtDNA as previously studied *Aplysina* species with 20 genes transcribed in the opposite direction comparing to *rnl* and the rest of the genes. No inversions were found in Heteroscleromorpha mtDNA: all genes had the same transcriptional polarity.

**Introns in *cox1*, *cox2*, and *rnl*.**   Seventeen *cox1* introns have been found in ten heteroscleromorph sponges sampled for this study: one in *Adreus fascicularis* (order Tethyida), one in *Acanthella acuta* (order Bubarida), two in each *Axinella polypoides* and *Axinella infundibuliformis* (order Axinellida), two in *Cliona varians* (order Clionaida), one in *Phakellia ventilabrum* (order Bubarida), two in *"Svenzea" flava* (classified as Scopalinida, but—based on our data— closely related to *Acanthella acuta* and *P. ventilabrum*), three in *Leiodermatium* sp., one in *Microscleroderma* sp., and two in *Myceliospongia araneosa* at positions. In addition, two *cox1* introns were found in the Verongimorpha *Thymosia* sp., doubling the number of previously known demosponge orders with mt-introns. Unexpectedly, we also found group II introns in two other mitochondrial genes: *cox2* of *Acanthella acuta* and *rnl* of *A. acuta*, *Dictyonella marsilii*, and another Dictyonellidae species (P0911). The *cox2* intron in *Acanthella acuta* contained an ORF most similar to the group II intron reverse transcriptase/maturase of a microalga *Ulva ohnoi*. The *rnl* intron in the two Dictyonellidae species was found in the same position as in the three placozoan species [67] and contained a large region that displayed high sequence similarity to a region within mt-lrRNA gene in those species. The structures and phylogenetic affinities of introns found in reported mt-genomes are being analyzed in a separate study (unpublished).

## Phylogenetic analyses based on a supermatrix of inferred amino acid sequences

Bayesian phylogenetic analysis based on concatenated amino-acid sequences derived from mitochondrial protein genes from 136 species of demosponges yielded a well-supported

consensus tree of demosponge relationships with the mean posterior probability support of bipartitions of 0.94 and 73% of bipartitions having maximum support Fig 2. Analysis of a subset of these species with six additional homoscleromorph species (outgroup) allowed us to position the root of demosponges on a branch between Keratosa + Verongimorpha and Heteroscleromorpha (Fig 1). Most heteroscleromorph orders proposed by Morrow and Cardenas [29] were recovered as monophyletic groups, although, species sampling was limited for most of them. Some cases where a particular species was not placed within its accepted order (*e.g.*, *Axinella corrugata*, *A. damicornis*, *Svenzea flava*, and *Topsentia ophiraphidites*) were probably because that previous order assignment was a misclassification (see [68] and Discussion). Within Heteroscleromorpha, we found support for five major clades (named here C0–C4).

- C0: Order Haplosclerida

- C1: Orders Spongillida, Scopalinida, and Sphaerocladina.

- C2: Orders Axinellida, Biemnida, Bubarida along with *Topsentia ophiraphidites* and *Petromica* sp.

- C3: Order Tetractinellida + *Myceliospongia*.

- C4: Orders Agelasida, Clionaida, Desmacellida, Merliida, Poecilosclerida, Polymastiida, Suberitida, and Tethyida.

The phylogenetic relationship among these clades was reconstructed as (C0(C1(C2(C3, C4)))), although the interrelationship among C2, C3, and C4 was only moderately supported.

## Phylogenetic analysis based on gene order data

Comparison of mitochondrial gene orders in heteroscleromorph species without major tRNA gene loss revealed that they shared with each other at least six gene boundaries, with the mean number of shared boundaries between a given species and the rest of heteroscleromorphs varying between ∼10 for *Plenaster craigi* and ∼29 for *Svenzea zeai* and Spongillida (freshwater sponges) (mean = 24.3). The gene orders were converted into a gene-based matrix, where the identities and transcriptional orientations of the upstream and downstream neighbors of each gene were recorded. The matrix was utilized for maximum likelihood (ML) analysis in RAxML-NG [59] and maximum parsimony (MP) analysis in TNT [58]. The results of these analyses (Fig 3, S2 Fig) generally support the Heteroscleromorpha relationships reconstructed from sequence data, including its subdivision into major clades and the phylogenetic position of *Myceliospongia araneosa*. However, four main discrepancies were found:

First, C1 clade (Spongillida, *Vetulina* sp. and Scopalinida) was not reconstructed as a monophyletic group. Instead *Vetulina* sp. and *Scopalina* sp. either grouped with Tetractinellida plus *Myceliospongia araneosa* or was a part of a polytomy at the base of the tree. The lack of support for C1 is not surprising, given that the gene order in Spongillida was inferred to be ancestral for the Heteroscleromorpha [26].

Second, phylogenetic position of the order Agelasida was unstable, as it either grouped with C3 + *Vetulina* sp. and *Scopalina* sp. (ML analysis) or its position was unresolved among the main clades. The rest of the C2 clade formed a sister group to all heteroscleromorphs but Haplosclerida.

Third, *P. craigi* grouped with *Agelas schmidti* and *Astrosclera willeyana* within Agelasida. As noted above, *P. craigi* has the most derived mitochondrial gene order, reflected also in the longest branch in the ML tree (Fig 3).

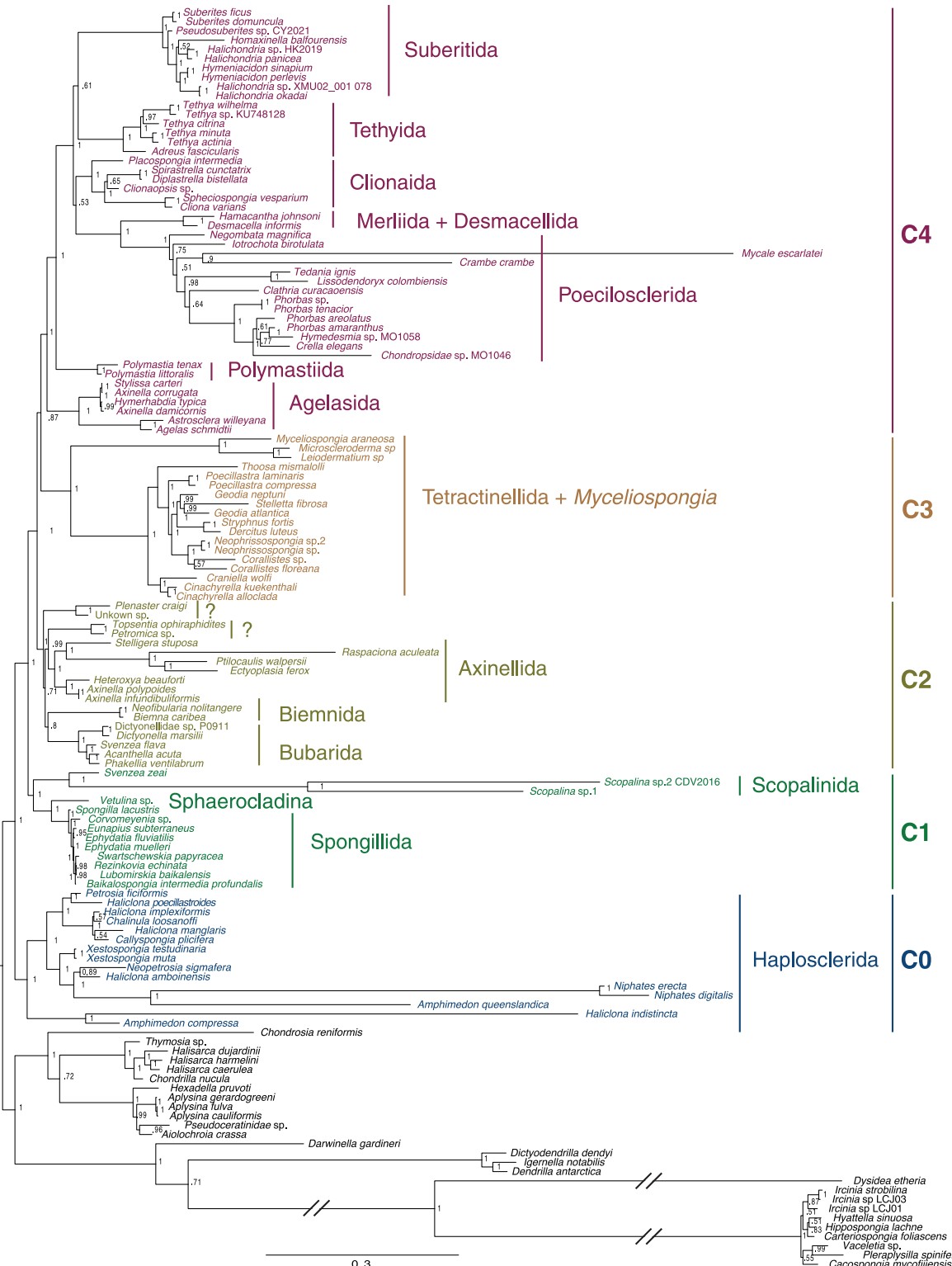

**Fig 2. Demosponge phylogenetic relationships based on translated mt-coding sequences.** Posterior majority-rule consensus tree was obtained from the analysis of concatenated mitochondrial amino acid sequences (3,634 positions) under the CAT+GTR+Γ model in the PhyloBayes-MPI program. The number at each node represents the Bayesian posterior probability. The branches marked by a broken line symbol are shown half of their actual lengths. Five major clades in Heteroscleromorpha are shown as C0–C4.

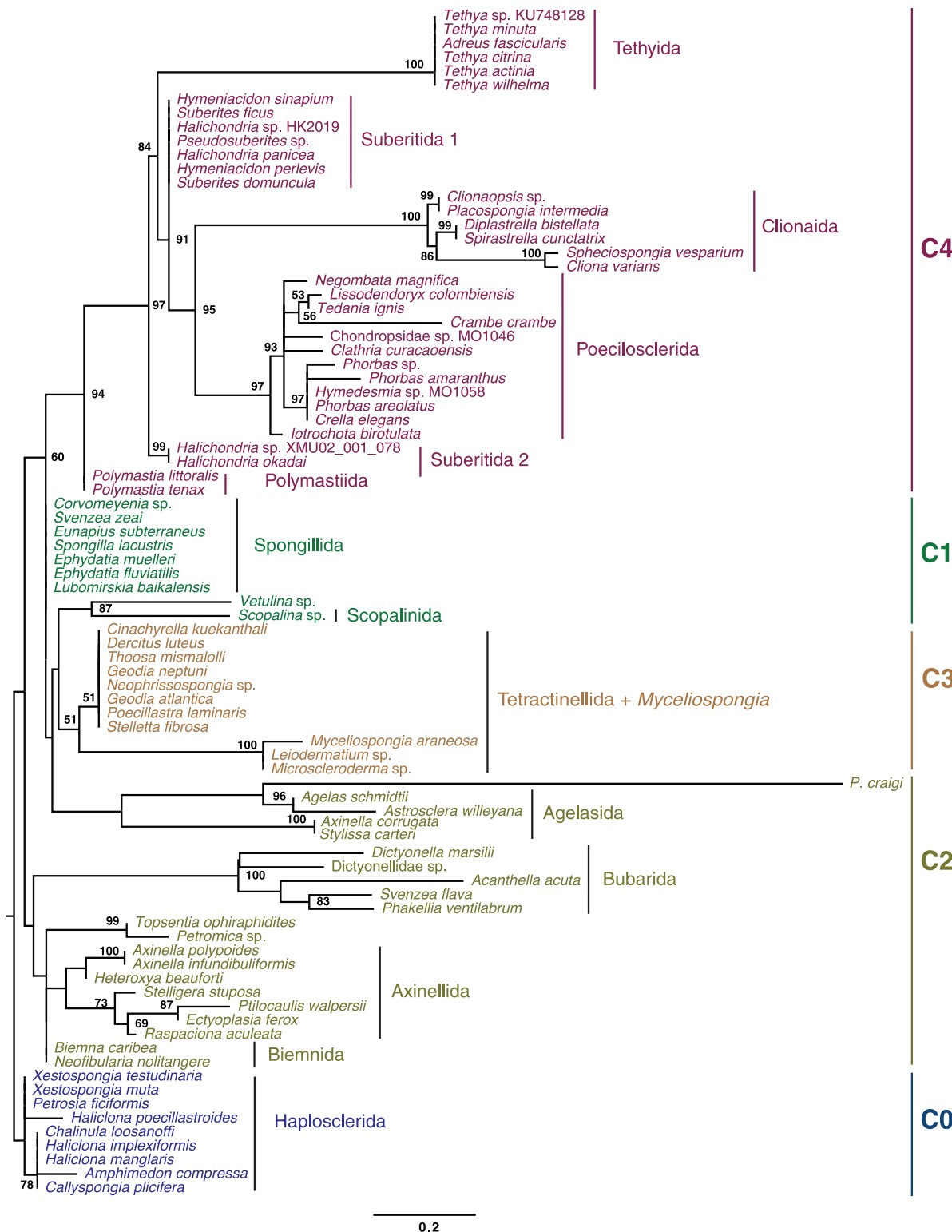

**Fig 3. Phylogenetic relationships among Heteroscleromorpha reconstructed from mitochondrial gene order data.** Gene boundaries were encoded as multistate characters based on identity and orientation of each gene. The matrix was used for Maximum Likelihood analysis under the MULTI13_MK model in RAxML-NG. Numbers above branches show the bootstrap support when >50%.

To check if any inconsistencies between the sequence-based and gene order-based phylogenies were due to saturation of phylogenetic signal in gene order data caused by frequent movement of tRNA genes, we repeated both ML and MP analyses using the multistate encoding based on closest major (protein or rRNA) genes (see [35] for details). Indeed, *P. craigi* grouped with C2 species in both of these analyses. However, the position of Agelasida was still unresolved and there was no support for the monophyly of Tetractinellida, although *Myceliospongia araneosa* still grouped with *Microscleroderma* sp., and *Leiodermatium* sp.

Finally, we visually identified several unique rearrangements associated with some large lineages of Heteroscleromorpha. Thus, we found that the sampled representatives of the orders Axinellida, Bubarida, Biemnida, along with *Topsentia ophiraphidites* and *Petromica* sp. shared a translocation of *trnR-nad4L* downstream of *rnl*. Interestingly, *trnR-nad4L* have also moved to this location in Haplosclerida, but as a part of a much larger genomic fragment that included three additional protein-coding genes: *nad1*, *nad2*, and *nad5* and several tRNA genes. We also found that orders Suberitida, Polymastiida, Poecilosclerida, and Clionaida shared a translocation of *trnY* and *trnI2* from a tRNA cluster between *nad1* and *nad2* to the region downstream of *rnl*. In addition, all Poecilosclerida species had a translocation of *cox2* into the gene junction between *nad5* and *rns*, while all Clionaida species had *trnV* moved immediately downstream of *cox1* from its conserved position between *trnG* and *rnl*.

## Molecular clock analysis

We used the same dataset as for the sequence-based phylogenetic analysis as well as the phylogenetic tree inferred in that analysis to estimate times of divergences among major lineages of demosponges (Fig 4, S3 Fig). Fixing the root of Demosponge phylogeny (between Heteroscleromorpha and Keratosa+Verongimorpha) between 541 and 515 MYA, we estimated that the divergence between Haplosclerida and the the rest of Heteroscleromorpha occurred between 534 and 488 MYA, followed by the split between C1 and C2+C3+C4 between 512–456 MYA, with the splits among C2–C4 between 494 and 417 MYA. Furthermore, most mean estimates for the divergences among orders of Heteroscleromorpha fall between the Ordovician and the Devonian, while those for the basal splits within the orders (that can be used as a proxy for the crown group) between Early Permian to Late Triassic. Several exceptions to this pattern have been found. First, we noted that the split between *Myceliospongia araneosa*, *Microscleroderma* sp., and *Leiodermatium* sp. with the rest of Tetractinellida occurred between 449 and 359 MYA, corresponding to the time of the origin of most orders. Second, the basal split within Axinellida (e.g., between Raspaillidae + Stelligeridae and Axinellidae) was estimated to occur at a similar time, between 438 and 324 MYA. By contrast, *Hamacantha johnsoni* (order Merliida) and *Desmacella informis* (order Desmacellida) were estimated to split only between 257 and 96 MYA and to diverge from Poecilosclerida between 363 and 266 MYA. Finally, we note here that the most basal divergence within Haplosclerida, estimated between 516 and 455 MYA, occurred at about the same time as the most basal divergence within the rest of Heteroscleromorpha (between C1 and C2+C3+C4). We suggest that these discrepancies—if confirmed by other datasets—can be used to re-define several orders in Heteroscleromorpha and to revise the taxonomic status of Haplosclerida (see Discussion).

## Discussion

### Assembling the new dataset of mtDNA data for demosponges

Despite rapid progress in sequencing technologies, sequence data remain scarce for many taxa of marine animals, including demosponges. When such data are available, they are often limited to partial sequences of individual genes, commonly those for nuclear rRNA

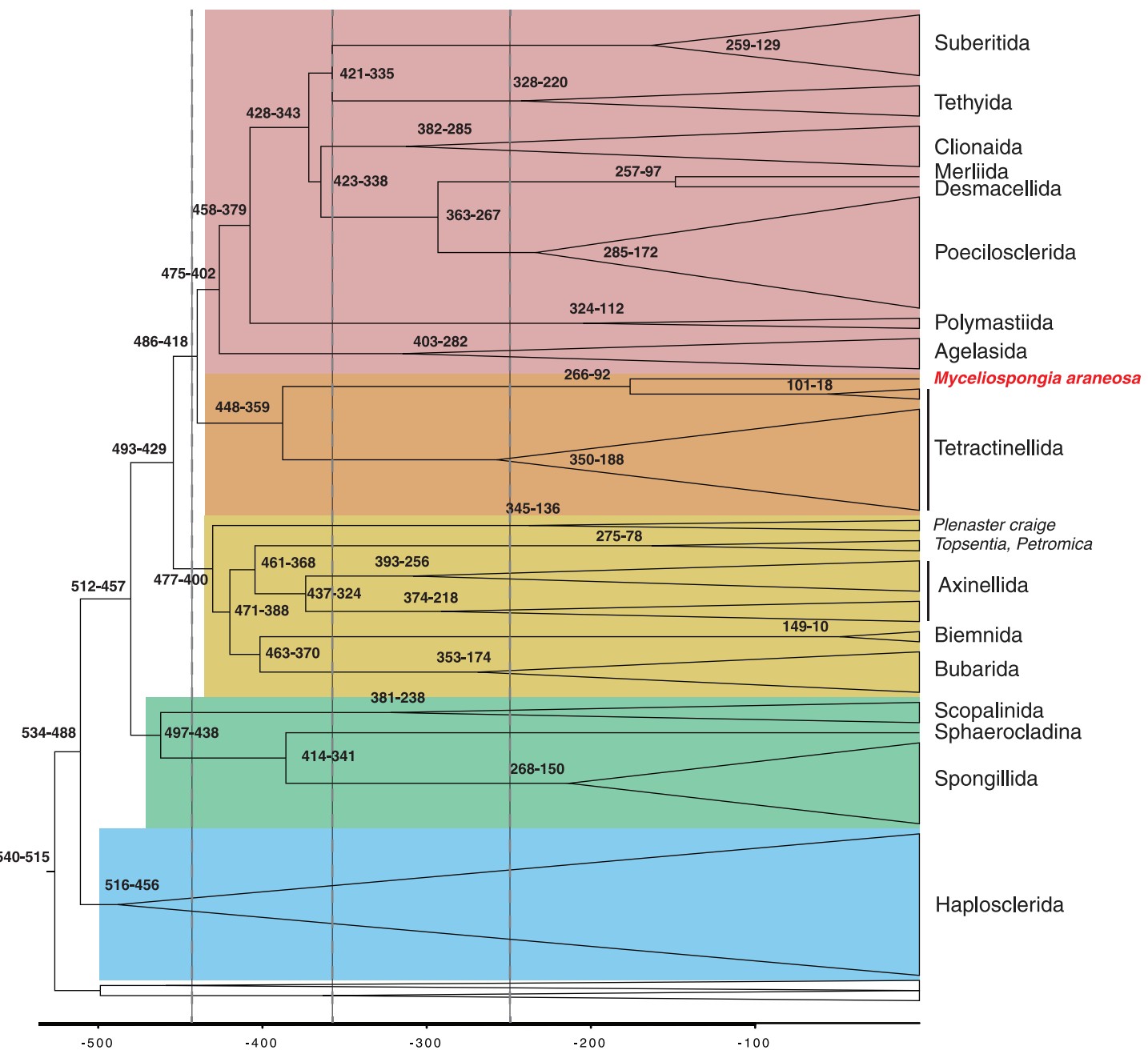

**Fig 4. Simplified time calibrated phylogeny of Demospongiae based on Bayesian analysis in PhyloBayes.** Only heterosleromorph orders are labeled. Numbers at internal nodes indicate their upper and lower age limits. The four large clades within Heteroscleromorpha are shaded in different colors. The vertical lines (left to right) correspond to the Ordovician-Silurian (444 MYA), the Devonian-Carboniferous (359 MYA), and the Permian-Triassic (252 MYA) boundaries. The full tree is shown in S3 Fig.

and/or mitochondrial *cox1*. While even these partial data have been instrumental for re-evaluation of demosponge relationships, larger datasets are needed to test and refine proposed phylogenetic hypotheses. Ultimately, a representative sampling of genomic loci should be a dataset of choice for a phylogenetic analysis. However, such datasets are available for only a few species of sponges. Furthermore, assembling and analyzing genomic

datasets present various challenges, especially in sponges, where contamination by foreign DNA is always a factor [69].

Here we compiled and analyzed a large dataset of mt-genomic sequences of demosponges, including 66 mtDNA sequences determined or assembled for this study. Importantly, we sampled all but one (Trachycladida) of the proposed orders within Heteroscleromorpha, the largest group of demosponges. We used this dataset to test phylogenetic relationships and reconstruct the timing of major splits within the group. Our results are largely consistent with and add resolution to phylogenetic studies based on single-gene data, compiled and synthesized by Morrow and Cárdenas [29]. For example, our sequence-based phylogenetic analysis not only groups Spongillida with Sphaerocladina (*Vetulina*)—as proposed by Redmond *et al.* [70]—and investigated in more details by Schuster *et al.* [71]—but also provides strong support for their association with Scopalinida (clade C1). Similarly, in addition to grouping orders Agelasida, Clionaida, Poecilosclerida, Polymastiida, Suberitida, and Tethyida (reported in [29]; clade C4 in our analysis), our results support a clade comprised of orders Axinellida, Biemnida, Bubarida along with *Topsentia ophiraphidites* and *Petromica* sp. (clade C2).

At the same time, there are a few differences between the results of this and previous molecular studies: *First*, Tetractinellida (C3) forms a sister group to C4 in our analysis rather than to Biemnida [72–74] or C2 (Axinellida, Biemnida, Bubarida) [29]. *Second*, within the clade C4, Tethyida groups with Suberitida rather than Clionaida [74] or Clionaida & Poecilosclerida [29]. *Third*, within C2, Bubarida is a sister group to Biemnida, rather than Axinellida [29, 74]. *Fourth*, *Topsentia ophiraphidites* and *Petromica* sp. (classified as Suberitida and *incertae sedis*, respectively in [29]) are placed within C2 in both sequence-based and gene-order based analyses and form a sister group to Axinellida in the former analysis (similar results are found in citepankey2022). Because there are no molecular data from the type species of either *Topsentia* or *Petromica*, the proper classification of these genera remains undetermined. However, it has been suggested that *Topsentia* could be a polyphyletic genus, with some species belonging to Axinellida and other to Suberitida [72].

Finally, we note that *Svenzea flava* (Lehnert & van Soest, 1999) groups with Dictyonellidae (Bubarida) species in both sequence and gene order analyses rather than with Scopalinida, which includes the type species *Svenzea zeai*. This finding is not too surprising, given that *S. flava* differs from other members of the genus in its skeletal arrangement and lacks characteristic granular cells and large embryos/larvae found in the type species [75]. Thus, in the future, *S. flava* will need to be renamed and reassigned to the family Dictyonellidae.

**Phylogenetic position of emerging sponge genomic model systems.** Our dataset included nine species of sponges for which high throughput DNA and/or RNA data were available and which, therefore, could be considered as emerging sponge model systems. Although the phylogenetic position of most of these species was as expected, two exceptions were found. First, *Stylissa carteri* (order Scopalinida) grouped closely with *Axinella corrugata* (order Axinellida) and both of them were placed as a sister group to *Axinella damicornis* (order Axinellida) and *Hymerhabdia typica* (order Agelasida) within Agelasida. Furthermore, the coding sequences between *Stylissa carteri* and *Axinella corrugata* were >99% identical, indicating a possible misidentification of the sponge. Second, the recently described abyssal sponge *Plenaster craigi*, grouped with the species from the orders Axinellida, Biemnida, and Bubarida in sequence-based phylogenetic reconstructions, but was not placed in any of these groups. Thus, it should not be consider a representative of Axinellida [76] and its phylogenetic position should be further investigated. Finally, we note that there was a significant level of cross-contamination between DNA sequence data from *Xestospongia testudinaria* and *Stylissa carteri* [16], such that both mitochondrial genomes can be assembled from either dataset.

## Are haplosclerids heteroscleromorphs?

The current demosponge taxonomy, which we followed in this article, places haplosclerid sponges within the subclass Heteroscleromorpha as the order Haplosclerida [77]. This fits the definition of the subclass as "Demospongiae with a skeleton composed of siliceous spicules which can be monaxons and/or tetraxons and when they are present, microsleres are highly diversified" [29]. However, the same definition was applied earlier to Heteroscleromorpha minus the order Haplosclerida, with the latter group considered as the fourth subclass of demosponges, Haploscleromorpha [22]. Does it matter if we call haplosclerid sponges an order or a subclass? According to Joe Felsenstein, a self-proclaimed founder of the It-Doesn't-Matter-Very-Much school of classification, "the delimitation of higher taxa is no longer a major task of systematics, as the availability of estimates of the phylogeny removes the need to use these classifications" [78]. Indeed, the position of the order Haplosclerida as the sister group to the rest of the Heteroscleromorpha—inferred in this and several previous studies—is consistent with both the four-subclasses classification system of Cardenas *et al.* [22] as well as the three-subclasses classification system of Morrow and Cardenas [29]. Nevertheless, we believe that the accepted status of Haplosclerida does influence our treatment of this taxon in two important ways. First, it changes the extent to which this large and diverse group of sponges is and will be represented in comparative studies, with a higher rank leading to a more thorough sampling. Second, it modulates our attention to several unusual features demonstrated by the group, including atypical patterns of evolution of ribosomal RNA genes [79, 80], "streamlined" nuclear genomes [21], and unusual content of the main skeleton-forming genes [81]. While the choice between the three vs. four subclasses classification of Demospongiae will—to a large extent—remain a subjective choice of the sponge taxonomic community, we note that the time estimate for the most basal divergence within Haplosclerida is comparable to that for the most basal divergence within the rest of Heteroscleromorpha (between C1 and C2+C3+C4) rather than the crown group divergences within other heteroscleromorph orders (see also citepankey2022). From this perspective, the classification of Haplosclerida as the fourth subclass of Demospongiae would be preferable

## Timing phylogenetic divergences

In addition to resolving phylogenetic relationships among clades of Heteroscleromorpha, we provide molecular clock estimates for the divergences among them. While several molecular clock studies using demosponge mt-genome data have been conducted [36, 38, 82], they utilized much smaller datasets of mitogenomic sequences. Furthermore, two of these studies [38, 82] used a suboptimal selection of the outgroup species, resulting in a likely erroneous phylogenetic reconstruction, which in turn biased molecular dating results (see below).

One major difficulty in molecular clock analysis of sponges is the scarcity and uncertainty of available calibration points. While the Paleozoic record of sponges is well established and extends to the Lowermost Cambrian (529–541 Ma) [83], the taxonomic affinities of Paleozoic sponges are often controversial. Nevertheless, the recent discovery of well-preserved fossils of crown-group demosponges [62] constrain the upper bound for the basal split in demosponges at 515 MY.

The Precambrian record of sponges is even more contentious. Although there have been numerous reports of Precambrian Porifera fossils, most of them are not substantiated [84]. Instead, the current argument for the Precambrian origin of demosponges is based primarily on the presence of fossil steroids (in particular 24-isopropylcholestanes and recently discovered 26-methylstigmastane) in the geological record before the end of the Marinoan glaciation (~635 MY ago) [85, 86]. However, the problem with both of these biomarkers is their erratic

distribution across modern demosponges, which at best can be interpreted as a result of multiple independent losses [86]. Such rampant loss makes it extremely difficult to estimate the origin of this biosynthetic pathway on a phylogenetic tree, especially given the sparse sampling of non-demosponge taxa [86]. Furthermore, the biosynthesis precursors of "sponge biomarkers" are found among Rhizaria, heterotrophic protists common in the ancient and modern oceans [87]. Thus, for the present study, we used the beginning of the Cambrian (541 MY) as the lower bound for the common ancestor of demosponges.

We used the origin of freshwater sponges (approximated by the split between the order Spongillida and *Vetulina* sp.) as the second calibration point for our analysis. The upper limit for the origin of freshwater sponges was defined by the oldest reported freshwater sponge fossils [64]. The lower bound was defined by a somewhat generic observation that species diversification in lakes prior to the Devonian was limited by low nutrient loads and high sediment loads [63]. There is substantial uncertainty with both of these estimates. First, the fossils we used to define the upper limit of the split predate most other known freshwater sponge fossils (mostly Jurassic and Cretaceous) by more than 100MY [88]. It is not clear if the lack of fossil freshwater sponges between the end of the Palaeozoic and the Jurassic is an artifact of preservation or if the colonization of the freshwater environment by sponges occurred more than once. Furthermore, although our intent was to estimate the origin of freshwater sponges, it is likely that the split between Spongillida and *Vetulina* happened in marine environment and thus preceded this event.

Our third calibration point was the origin of the crown group Baikal sponges (defined as the split between *Baikalospongia intermedia profundalis* and *Lubomirskia baikalensis*) and placed between 30 and 6 MY. The lower bound for this estimate is based on the Lake Baikal age (estimated between 25–30 MY), and lack of Lubomirskiidae spicules in early Tertiary sediments of the Tunkinskaya land basin (approximately Oligocene or 23–33 MYA) [65]. The upper bound is based on the analysis of Baikalian bottom sediment samples conducted during Baikal Drilling Project, which revealed well-formed spicules of several species belonging to all four genera of the Lubomirskiidae family in deposits corresponding to 6,50—4,75 MYA [65].

Given the substantial uncertainties in the calibration points and a large variance associated with molecular clock analysis, divergence times estimates for sponges need to be treated as only rough approximations. Nevertheless, it is illuminating to see that many proposed orders of demosponges have an ancient (Mid-Paleozoic) origin. By contrast, the origin of crown groups within most of them corresponds to the Late-Paleozoic or Mesozoic. These observations suggest that estimated divergence times could be taken into consideration when demosponge classification is revised and/or when alternative classification schemes are considered.

## Phylogenetic position of *Myceliospongia araneosa*

*Myceliospongia araneosa*, the only described species in the genus *Myceliospongia* Vacelet & Perez, 1998, is one of the most unusual demosponges in its anatomy and cytology. The sponge is known from a single cave (the "3PP" cave near La Ciotat (43˚09.47'N—05˚36.01'E)) in the Mediterranean Sea, where it grows on vertical or overhanging walls in a cool stable environment. *M. araneosa* has an encrusting 'body' of up to 25 cm diameter and 1 mm thick surrounded by a reticulation of filaments from as little as 5 $\mu m$ in diameter. These filaments form an extensive network around the body that can grow over and under other marine life. The sponge body has a reduced aquiferous system and a low number of small choanocyte chambers. Neither spicules nor spongin fibers are present, and collagen fibrils do not form bundles or thick condensations. *M. araneosa* shows a remarkable uniformity of cell types, with the mesohyl cells being poorly differentiated [89]. Because of the highly unusual organization,

which offered little clues as to its affinities to other sponges, *Myceliospongia* was placed in Demospongiae, *incertae sedis* [90].

Phylogenetic analyses based both on sequence and gene order data unequivocally placed *Myceliospongia* within Tetractinellida, and as the sister group to *Microscleroderma* sp. + *Leiodermatium* sp. The split between these taxa and the rest of Tetractinellida has been estimated to occur 456—385 MYA, a time more consistent with inter- rather than intra-order divergence (see above). Interestingly, both *Microscleroderma* sp. and *Leiodermatium* sp. are "lithistid" sponges characterized by the presence of articulated choanosomal spicules called desmas. It would be interesting to know whether *M. araneosa* also evolved from a lithistid ancestor, which would predicate losses of both choanosomal and ectosomal skeletons. Overall, losses of mineral skeleton are rare although not unprecedented in Heteroscleromorpha (*e.g.*, Haplosclerida: *Dactylia*), but can be regarded as a dominant feature in evolution of subclasses Keratosa and Verongimorpha [91]. Independent losses of mineral skeleton have also occurred in various clades of Homoscleromorpha (both within Oscarellidae and also Plakinidae) [92].

## Promise, pitfalls, and limitations of Porifera phylomitogenomics

Mitochondrial DNA (mtDNA) in general and animal mtDNA in particular has been a popular and a widely used molecular marker in phylogenetic studies [93]. This popularity is due to a combination of several features that facilitate the acquisition of mtDNA, including its typically circular organization, large copy number per cell, and a high proportion of coding sequences, with those that simplify analysis, such as absence of paralogs and introns, stable gene content and gene order. Phylogenetic analyses of sponge mt-genomes additionally benefit from relatively low rates of sequence evolution and a more homogeneous nucleotide composition (at least within individual classes of sponges) [94]. Indeed, previous studies by our and other groups have demonstrated the utility of mitogenomic datasets for reconstructing phylogenetic relationships within the phylum Porifera and the overall congruence among the results of phylomitogenomic analyses with those based on 18S, 28S, and transcriptomic datasets [26, 36, 42, 95].

Nevertheless, as with any dataset, phylogenetic analysis based on mitogenomic data can fail to reconstruct the true evolutionary history of organisms. The primary challenges faced by a phylogenetic analysis are erroneous (non-phylogenetic) signal due to stochastic noise and/or systematic bias [96] as well as incongruences between gene trees and the species tree due to non-orthology of sequences, horizontal gene transfer, or incomplete lineage sorting [97]. Although large genomic (including mitogenomic) datasets generally benefit from the increase in the ratio of the true phylogenetic signal (which adds across genes) to random noise (which does not), they do not solve the problem of non-phylogenetic signal caused by systematic biases, which also accumulates [98]. Similarly, while paralogs are rare in mtDNA and duplicated sequences are quickly eliminated, potential problems exist both with horizontal gene transfer (primarily through mtDNA introgression) [99] and incomplete lineage sorting [100].

The two well understood factors that can contribute to systematic biases in phylogenomic analyses are heterogeneous sequence compositions and variable evolutionary rates across taxa [96]. While nucleotide composition of mt-genomes is relatively uniform across demosponges (and also similar in homoscleromorphs) [101], it is markedly different between demosponges, glass sponges, and calcareous sponges [102–104]. Because commonly used models in phylogenetic analysis assume a stationary nucleotide/amino-acid composition across the phylogenetic tree, inter-class comparisons of sponges based on mt-sequences can be error-prone. It is important to note that both glass and calcareous sponges also utilize more derived mitochondrial genetic codes, leading to additional differences in patterns of sequence evolution (reviewed in [94]).

In contrast to sequence composition, most standard phylogenetic analyses place no constraints on rates of sequence evolution among branches of a phylogenetic tree. Nevertheless, variable evolutionary rates can result in "long branch attraction" (LBA)—an artificial grouping of taxa with higher rates of sequence evolution [105]. This phenomenon occurs because models of sequence evolution typically underestimate its complexity and thus undercount the number of changes occurring along long branches [106]. One common example of LBA is grouping of fast-evolving in-group taxa with an outgroup, the latter, by its nature, forming a long branch on a phylogentic tree. Thus, the choice of the model and the choice of outgroups are critical for phylogenomic analysis.

The systematic biases discussed above can explain unusual results obtained in two recent studies of demosponge relationships that utilized mitogenomic datasets [38, 82]. First, in the paper by Schuster et al. [82] a time-calibrated phylogeny of Demosponges shows both Haplosclerida and Spongillida nested deep within Heteroscleromorpha with Tetractinellida forming the most basal divergence with the rest of heteroscleromorphs. These unusual placements of Haplosclerida, Spongillida, and Tetractinellida are incongruent with most previous phylogenetic reconstructions as well as with the standard ML and Bayesian trees obtained in the same study but presented only as supplementary data. It is likely that this result is a combination of two factors: the choice of the outgroup (order Dictyoceratida) and the use of molecular clock analysis to co-infer phylogeny. Dictyoceratida are known to form an extremely long branch on the demosponge phylogenetic tree [26], which likely attracts other longer branches to the base of the tree. The suboptimal choice of the outgroup is likely exacerbated by constraints (a prior distribution) placed by molecular clock analysis on the rates of sequence evolution needed for time estimation.

Second, a more recent paper by Plese et al. [38] reconstructed unconventional relationships among subclasses of demosponges, with Verongimorpha grouping with Heterosleromorpha rather than Keratosa. Again, the choice of the outgroup—class Hexactinellida—was unfortunate. Although the accepted view of sponge relationships places Demospongiae + Hexactinellida and Homoscleromorpha + Calcarea as sister groups (reviewed in [107]), mt-genome evolution is more similar in Demospongiae and Homoscleromorpha, whereas the other two classes accumulated multiple idiosyncrasies in mtDNA [94]. In particular, Hexactinellida display patterns of sequence evolution very different from those in demosponges, utilize a distinct mitochondrial genetic code, and have higher rates of mt-sequence evolution [102]. Because Keratosa are also represented by a long branch in a mitogenomic analysis, its inferred position is likely the result of the LBA between this taxon and Hexactinellida.

While one might argue that any phylogeny is just a hypothesis of evolutionary relationships, and that having diverse hypotheses could stimulate discussion and expedite progress, this is often not the case. Conflicting phylogenies derived from the same or similar datasets can diminish confidence in molecular phylogenetics, particularly among biologists who are not specialists in the field. This lack of agreement can also complicate efforts to combine information from multiple phylogenetic datasets, as attempted by projects like OpenTree [108] and TimeTree [109]. Moreover, when inaccurate phylogenies propagate through the biological literature, they can result in misinterpretations of comparative data. Hence, it is crucial to address known issues in phylogenetic reconstruction, both during dataset construction and the actual inference process [69].

## Conclusion

In this study, we acquired 64 complete or nearly complete and six partial mt-genome sequences, more than doubling the number of demosponge species available for

phylomitogenomic analysis. We compiled these new data with pre-existing records to build a dataset of 136 demosponge species and to test the high-rank phylogeny in the class Demospongiae, in particular the order composition of the subclasses and the relationships between subclasses. Our phylogenetic reconstruction is consistent with a subdivision of Demospongiae into either three subclasses (Verongimorpha, Keratosa, Heterosclermorpha) or four subclasses (Verongimorpha,Keratosa, Heterosclermorpha, Haplosclermorpha). However, we argue that classifying the current heterosleromorph order Haplosclerida as a fourth subclass of Demospongiae provides better agreement with estimated times of diversification and observed genomic diversity within this group. We confirmed that Heteroscleromorpha forms the sister group to the rest of the demosponges (Haploscleromorpha (Keratosa, Verongimorpha)) and that a recent result claiming closer relationships between Heteroscleromorpha and Verongimorpha is likely an artifact of long branch attraction between Keratosa and the selected outgroup. Analysis of molecular data from *Myceliospongia araneosa*, a highly unusual skeleton-lacking demosponge, surprisingly placed it within the heteroscleromorph order Tetractinellida, as a sister taxon to the highly skeletonized *Microscleroderma* sp. and *Leiodermatium* sp. In general, we argue that mt-genomes provide an informative dataset for studying demosponge phylogenetic relationships but that one must be careful with the choice of outgroups and models of sequence evolution. Finally, we created a website https://lavrovlab.github.io/Demosponge-phylogeny that contains all the data and the trees presented in this paper and that we plan to update as additional mitogenomic data from demosponges become available.

## Supporting information

**S1 Fig. Rooted phylogenetic tree of Demospongiae.** Posterior majority-rule tree was obtained from the analysis of concatenated mitochondrial amino acid sequences (3,633 positions) under the CAT+GTR+$\Gamma$ model in the PhyloBayes-MPI program. Mitochondrial coding sequences from nine species of Homoscleromorpha were added to those of demosponges used for the analysis presented in Fig 2 and the resulting dataset was filtered with CD-Hit to remove sequences with >95% identity. The root was placed between Demospongiae and Homoscleromorpha.
(PDF)

**S2 Fig. Maximum Parsimony reconstruction of Demospongiae relationships based on mitochondrial gene boundaries.** Strict consensus of 12800 most parsimonious trees is shown with support values based on 200 bootstrap replicates. All analyses were conducted in TNT v. 1.5 using the "new technology" search options.
(PDF)

**S3 Fig. Time calibrated phylogeny of Demospongiae based on Bayesian analysis in PhyloBayes.** Numbers at internal nodes indicate their upper and lower age limits.
(PDF)

## Acknowledgments

We are grateful to our colleagues Steven Cook, Alexander Ereskovsky, April and Hill, Gisele Lôbo-Hajdu, Michael Nickel, Julie Reveillaud, and Gert Wörheide for samples contributed to this project; Malcolm Hill, Jenna Moore, Bernard Picton, and John Reed for help with specimen collection; Andrzej Pisera and Sven Zea for help with species identification. We also thank the following former graduate and undergraduate students in the Lavrov Lab for their help with the molecular work: Andrea Bekic, Jenessa Filler, Ehsan Kayal, Philip Lange, Katrina Lutap, Benjamin Sheller, Xiujuan Wang, and Katherine Wilson. We are grateful to Nicole

Boury-Esnault and three anonymous reviewers for their careful reading of the previous versions of the ms. and many thoughtful suggestions.

## Author Contributions

**Conceptualization:** Dennis V. Lavrov, Robert W. Thacker.

**Data curation:** Dennis V. Lavrov, Robert W. Thacker.

**Formal analysis:** Dennis V. Lavrov.

**Funding acquisition:** Dennis V. Lavrov, Manuel Maldonado, Robert W. Thacker.

**Investigation:** Dennis V. Lavrov, Maria C. Diaz, Manuel Maldonado, Shirley A. Pomponi, Robert W. Thacker.

**Project administration:** Dennis V. Lavrov, Robert W. Thacker.

**Resources:** Dennis V. Lavrov, Maria C. Diaz, Manuel Maldonado, Christine C. Morrow, Thierry Perez, Shirley A. Pomponi.

**Supervision:** Dennis V. Lavrov.

**Visualization:** Dennis V. Lavrov.

**Writing – original draft:** Dennis V. Lavrov.

**Writing – review & editing:** Dennis V. Lavrov, Maria C. Diaz, Manuel Maldonado, Christine C. Morrow, Thierry Perez, Shirley A. Pomponi, Robert W. Thacker.

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
