## [Decision Letter · Decision Letter 0]

7 Sep 2023

PONE-D-23-16972Phylomitogenomics bolsters the high-level classification of Demospongiae (phylum Porifera)PLOS ONE

Dear Dr. Lavrov,

Thank you for submitting your manuscript to PLOS ONE. After careful consideration, we feel that it has merit but does not fully meet PLOS ONE’s publication criteria as it currently stands. Therefore, we invite you to submit a revised version of the manuscript that comprehensively addresses the points raised during the review process.

We look forward to receiving your revised manuscript.

Kind regards,

Michael Schubert

Academic Editor

PLOS ONE

Reviewers' comments:

Reviewer's Responses to Questions

**Comments to the Author**

1. Is the manuscript technically sound, and do the data support the conclusions?

Reviewer #1: Yes

Reviewer #2: Yes

Reviewer #3: Yes

2. Has the statistical analysis been performed appropriately and rigorously? 

Reviewer #1: Yes

Reviewer #2: Yes

Reviewer #3: Yes

3. Have the authors made all data underlying the findings in their manuscript fully available?

Reviewer #1: Yes

Reviewer #2: Yes

Reviewer #3: Yes

4. Is the manuscript presented in an intelligible fashion and written in standard English?

Reviewer #1: Yes

Reviewer #2: Yes

Reviewer #3: Yes

5. Review Comments to the Author

Reviewer #1: I congratulate the authors on getting this huge dataset together and writing it up! I'd also like to thank them for submitting such a polished version for review, as it makes my job easy. The discussion, especially, was very nice. I love the authors honest discussions about doubt and uncertainty regarding fossil calibration. Great work!

The dataset presented here, and its analysis, are undoubtedly valuable to the sponge community. The raw data will be used by many, and the tree will be thought provoking for a wide swath of evolutionary biologists. This value is also increased by the recent papers by Schuster et al. and Plese et al. that require correction, as pointed out (fairly) by the authors in the discussion. I would suggest the manuscript be accepted with only minor revisions required. My comments are very minor.

One point that struck me, however, was the omission of the recent paper by Pankey et al. 2022: https://www.nature.com/articles/s41559-022-01712-3 (and for the record: I'm not an author on that paper). The Pankey paper has time-calibrated phylogenies based on multiple loci, and is thus the most up-to-date and large-scale phylogeny available. The authors need to cite the work, and compare their results to Pankey. I think, for the most part, the results are very congruent -- even in terms of estimate divergence times, amazingly! -- and the papers will support each other very well (and add support to Lavrov et al.'s claims that the Plese & Schuster results are erroneous). The one possible exception might be that the established orders appear to be more problematic with the larger species sampling of Pankey -- though it is hard to tell how much data is available for any given species in their analysis, and they discuss their phylogenetic results in much less detail than Lavrov (indeed, barely at all!).

Additional, exceedingly minor points:

I believe one of the author's names is misspelled

Intro: line 52: you mean former not latter, right?

You list voucher numbers in table 1, but it is unclear if these are actually archived in a permanent collection (and I really hope they are, since morphological data supporting the species IDs is not provided here, and some could be misidentified!). The authors need to explain what museum collections each of these numbers refers to, as no one can find " 28-V-91-2-001", for example, without knowing where to look. If some of these are vouchered in the author's labs instead of museums, I highly encourage the authors to voucher them in a more stable place.

The Plenaster species name is misspelled in the time-calibrated tree, I think

Fig 4 caption mentions superorders...? Maybe proposed superorders? I don't know about any designated superorders.

Is fig S1 referenced in the paper? If it is, I missed where.

Reviewer #2: I congratulate the authors on this long awaited fine piece of work. I found no conceptual issues to remark upon, but I confess I am not proficient on the details of the analyses undertaken. I can feel the underlying anxiety of wanting to turn big pieces of the current classification upside down, but authors were correctly conservative in this respect, which is expected as long as the Linnean typological classification rules our relation with the diversity of life. Said all this, please check the attached revised MS carfully for a series of typos highlighted. Also, I see a need for a carefull review of the references too, as many are abbreviated, while others are not.

Reviewer #3: The manuscript by Lavrov and collaborators presents very interesting information regarding the systematic position of major demosponge orders recovered by the use of mitochondrial genomes, which is fundamental for understanding Porifera evolution. The authors have done a fantastic job sampling rare species and orders, and assembling a great number of new mitochondrial genomes, which are fundamental to understand sponge phylogeny. I think the manuscript deserves publication after major revision though, especially because there are some omissions of important results that should be discussed in depth. Also, figure readability is crucial to make science accessible, and I would suggest improving the quality of the figures presented here.

Introduction

There seems to be an incongruence in the number of datasets sequenced and assembled: 64 sequenced and 6 assembled from existing data in the introduction, while it says 59 sequenced and 5 assembled from existing datasets. Please ensure these numbers are correct in the manuscript.

Methods

I am afraid that the Table 1 provided is not sufficient to comply with the current standard metadata details necessary for reproducibility and open science. Please provide accurate location details, with longitude and latitude at least.

In addition, there is one species coded as Unknown sp. Although I understand the importance of sequencing, the inclusion of completely undetermined species in phylogenetic trees with such relevance for sponge systematics should be avoided. Please provide some better taxonomic identification of the sponge, which I believe would not be extremely difficult given that among the authors are some of the best sponge taxonomists of the field.

Please provide details for MAFFT alignment parameters.

Mitochondrial genes are highly conserved across demosponges in general, so it seems unnecessary to use Gblocks to produce alignments. What was the rationale behind this? Did the authors try a non-gblocked alignment to compare the convergence?

Results

The finding of multi-chromosomic mitochondrial genomes in demo sponges is highly important and novel, and deserved more attention. There is not sufficient information to understand the specific features of the mt genome of Mycale escalatei. I understand this would be in line with other preliminary data, but since that is not provided I suggest discussing here other possibilities for the lack of circularity (incompleteness, repetitive sequences, etc).

No information is provided for the position of the introns in the mt genes because it says they are being analysed in a separate study. Perhaps the phylogenetic affinities of such introns can be left for this separate study, but if the presence of introns is reported here, I would strongly encourage the authors to provide details somewhere for their positions.

Discussion

Lines 395-399: The recovery of a clade formed by Vetulina and Spongillida was also previously found by Plese et al. 2021. These lack of discussion with Plese et al. 2021 seems to be pervasive along the discussion, where it is only cited to discuss the unconventional relationships found there, which the authors claim that are likely to be found because unfortunate selection of outgroups. Although I don’t particularly agree with the view of the authors regarding these unfortunate outgroups, this could very well have been the case. However, that does not invalidate the entire results of Schuster et al. and Plese et al., which in turn found many of the same relationships found in the present study. One may wonder if there was a deliberate intention of not discussing the similar results found by these studies. For instance, the authors highlight the “unexpected” phylogenetic position of Axinella damicornis, although that was already placed within Agelasida by Plese et al. 2021. I would suggest discussing fairly and honestly the results of other researchers, giving credit where it is deserved, because it does not diminish the enormous relevance and importance of the present study.

The main figure of the paper (Figure 2) is extremely difficult to follow and in my opinion, the quality is not sufficient for publication. Same for Figure 3. I suggest editing the figure to make the tree more readable and provide a summary tree with the relationships of the orders.

Other minor comments:

The name of the last author is not spelled correctly.

6. PLOS authors have the option to publish the peer review history of their article (what does this mean?). If published, this will include your full peer review and any attached files.

Reviewer #1: No

Reviewer #2: No

Reviewer #3: No

---

## [Author Response · Author response to Decision Letter 0]

17 Oct 2023

Please see our response in the Response_to_Reviewers file.

---

## [Decision Letter · Decision Letter 1]

31 Oct 2023

PONE-D-23-16972R1Phylomitogenomics bolsters the high-level classification of Demospongiae (phylum Porifera)PLOS ONE

Dear Dr. Lavrov,

Thank you for submitting your manuscript to PLOS ONE. After careful consideration, we feel that it has merit but does not fully meet PLOS ONE’s publication criteria as it currently stands. Therefore, we invite you to submit a revised version of the manuscript that addresses the points raised during the review process. More specifically, please address the quality issue of the figures raised by reviewer #3 and please remove the paragraph as suggested by reviewer #3, as it does not pertain to the discussion of the results obtained in this study.

We look forward to receiving your revised manuscript.

Kind regards,

Michael Schubert

Academic Editor

PLOS ONE

Journal Requirements:

Reviewers' comments:

Reviewer's Responses to Questions

**Comments to the Author**

1. If the authors have adequately addressed your comments raised in a previous round of review and you feel that this manuscript is now acceptable for publication, you may indicate that here to bypass the “Comments to the Author” section, enter your conflict of interest statement in the “Confidential to Editor” section, and submit your "Accept" recommendation.

Reviewer #1: All comments have been addressed

Reviewer #3: (No Response)

2. Is the manuscript technically sound, and do the data support the conclusions?

Reviewer #1: Yes

Reviewer #3: Yes

3. Has the statistical analysis been performed appropriately and rigorously? 

Reviewer #1: Yes

Reviewer #3: Yes

4. Have the authors made all data underlying the findings in their manuscript fully available?

Reviewer #1: Yes

Reviewer #3: Yes

5. Is the manuscript presented in an intelligible fashion and written in standard English?

Reviewer #1: Yes

Reviewer #3: Yes

6. Review Comments to the Author

Reviewer #1: (No Response)

Reviewer #3: I believe the authors have accommodate almost all changes suggested to improve their manuscript. However, I still believe there are 2 figures without the quality necessary to fully grasp their results (2 and 3). Also, I don't fully understand their last paragraph about the results of other researchers "degrading confidence in phylomitogenomics and the field of molecular phylogenetics in general". Conflicting phylogenies are everywhere, and their inherent to the field, given the nature of the field, relying on parameter, taxa, and approach choices. There is no need to be aggressive or "nasty" to other researchers just because your results and approach disagree with them. I suggest removing the paragraph, and other than that I have no more comments.

7. PLOS authors have the option to publish the peer review history of their article (what does this mean?). If published, this will include your full peer review and any attached files.

Reviewer #1: No

Reviewer #3: No

---

## [Author Response · Author response to Decision Letter 1]

12 Nov 2023

Thank you for your assistance with our manuscript, titled "Phylomitogenomics bolsters the high-level classification of Demospongiae (phylum Porifera)." We have modified figures 2, 3, and 4 in the manuscript and believe that their quality has improved. We have also modified the last paragraph of the Discussion. We note that in this paragraph we intended to provide a concluding message for readers to take home regarding the strength and importance of mitochondrial data, a message that is pertinent to our study. We did not intend to criticize other studies (so we removed the reference to Fig1 in [19]), but to address the impact of conflicting phylogenies derived from closely related mitochondrial datasets. In response to the comment by the reviewer and to further clarify our point of view, we have rephrased the paragraph as follows:

“While one might argue that any phylogeny is just a hypothesis of evolutionary relationships, and that having diverse hypotheses could stimulate discussion and expedite progress, this is often not the case. Conflicting phylogenies derived from the same or similar datasets can diminish confidence in molecular phylogenetics, particularly among biologists who are not specialists in the field. This lack of agreement can also complicate efforts to combine information from multiple phylogenetic datasets, as attempted by projects like OpenTree [108] and TimeTree [109]. Moreover, when inaccurate phylogenies propagate through the biological literature, they can result in misinterpretations of comparative data. Hence, it is crucial to address known issues in phylogenetic reconstruction, both during dataset construction and the actual inference process [69]”

Sincerely Yours,

Dennis Lavrov

---

## [Editor Report · Decision Letter 2]

15 Nov 2023

Phylomitogenomics bolsters the high-level classification of Demospongiae (phylum Porifera)

PONE-D-23-16972R2

Dear Dr. Lavrov,

We’re pleased to inform you that your manuscript has been judged scientifically suitable for publication and will be formally accepted for publication once it meets all outstanding technical requirements.

Kind regards,

Michael Schubert

Academic Editor

PLOS ONE

---

## [Editor Report · Acceptance letter]

21 Nov 2023

PONE-D-23-16972R2 

Phylomitogenomics bolsters the high-level classification of Demospongiae (phylum Porifera) 

Dear Dr. Lavrov:

I'm pleased to inform you that your manuscript has been deemed suitable for publication in PLOS ONE. Congratulations! Your manuscript is now with our production department. 

Kind regards, 

on behalf of

Dr. Michael Schubert 

Academic Editor

PLOS ONE